

# Structural complexities and tectonic barriers controlling recent
# seismic activity of the Pollino area (Calabria-Lucania, Southern Italy)
# - constraints from stress inversion and 3D fault model building
Daniele Cirillo[1-2]*, Cristina Totaro[2-3], Giusy Lavecchia[1-2], Barbara Orecchio[2-3], Rita de Nardis[1-2], Debora
Presti[2-3], Federica Ferrarini[1-2], Simone Bello[1-2] and Francesco Brozzetti[1-2]*
[1] Università degli studi "G. d'Annunzio" Chieti-Pescara, DiSPUTer, via dei Vestini 31, 66100 Chieti, Italy.
[2] CRUST Centro inteRUniversitario per l'analisi SismoTettonica tridimensionale, Italy.
[3] Università degli studi di Messina, Dipartimento di Scienze Matematiche e Informatiche, Scienze Fisiche e Scienze della Terra
-Viale F. Stagno D'Alcontres, 98166, Messina, Italy
*Correspondence to: d.cirillo@unich.it; francesco.brozzetti@unich.it
**Abstract.** The integration of field geology and high-resolution seismological data allowed us to reconstruct the 3D Fault Model
of the sources which gave rise to the 2010-2014 Pollino seismic sequence.
The model is constrained at the surface by structural geological data which provide the true attitude of the single faults and
their cross-cut relationships. At depth, the fault geometry was obtained using the distributions of selected high-quality relocated
hypocenters. Relocations were carried out through a non-linear Bayloc algorithm, followed by the double-difference relative
location method HypoDD, applied to a 3D P-wave velocity model.
Geological and seismological data converge in describing an asymmetric active extensional fault system characterized by an
E to NNE-dipping low-angle detachment, with its high-angle synthetic splays, and SW- to WSW-dipping, high-angle antithetic
faults.
The cluster of hypocenters and the peculiar time-space evolution of the seismic activity highlight that two sub-parallel WSW-
dipping seismogenic sources, namely the Rotonda-Campotenese and Morano-Piano di Ruggio faults activated during the
seismic crisis.
By applying to the activated structures the appropriate earthquake-scaling relationships, based on fault length and fault area,
we infer that the maximum expected magnitudes calculated using the fault area are the more reliable. We estimated $M_w$=6.4
for the Rotonda-Campotenese and $M_w$=6.2 for the Morano-Piano di Ruggio deducing that both the faults did not release their
seismic potential during the 2010-2014 seismic sequence.
The size of the activated patches, reconstructed by projecting on the 3D seismogenic fault planes the early aftershocks of the
seismicity clusters, are consistent with the observed magnitude of the associate strongest events.


Finally, we point out that the western segment of the Pollino Fault, despite not being presently active, acts as a barrier to the
southern propagation of the Rotonda-Campotenese and Morano-Piano di Ruggio faults, limiting their dimensions and
seismogenic potential.

## 1 Introduction

In recent years, the reconstruction of 3D Fault Models (hereinafter referred to as 3DFM) of potentially seismogenic structures
has become an increasingly practiced methodology in the seismotectonic analysis of regions undergoing active deformation
(SCEC, 2021; Lavecchia et al., 2017; Castaldo et al., 2018; Di Bucci et al., 2021). The techniques to obtain the 3DFM integrate
all the available surface ad subsurface data and allow to reconstruct the 3D geometry of seismogenic structures. In particular,
detailed structural-geological data are used to define the geometry of the active faults at the surface whereas high-quality
geophysical data are needed to constrain the shape of the sources at depth. If these conditions are met, 3DFM reconstruction
allows determining the spatial relationships and the interactions between adjacent sources and identifying any barriers
hampering at depth the propagation of the coseismic rupture. Moreover, such an approach leads to estimate with great accuracy
the area of the seismogenic fault and the associated expected magnitude.
In Italy, reconstruction of 3DFM could give important achievements in the Apennine active extensional belt which is affected
by significant seismic activity (ISIDe, 2007; Rovida et al., 2020). This belt consists of ~NW-SE striking Quaternary normal
fault systems, and the related basins, located just west or within the culmination zone of the chain (Calamita et al., 1992;
Brozzetti and Lavecchia, 1994; Lavecchia et al., 1994, 2021; Barchi et al., 1998; Cinque et al., 2000; Brozzetti, 2011; Ferrarini
et al., 2015). Its structural setting is very complicated due to a polyphase tectonic history characterized by the superposition of
Quaternary post-orogenic extension on Miocene-Early Pliocene folds and thrusts and on Jurassic-Cretaceous sin-sedimentary
faults (Elter et al., 1975; Ghisetti and Vezzani, 1982, 1983; Lipmann-Provansal, 1987; Patacca and Scandone, 2007; Mostardini
and Merlini, 1986; Vezzani et al., 2010, among others).
Since the beginning of the Pleistocene, regional normal fault systems dissected the contractional structures with displacements
that locally reach some km. Over time, detailed structural geological studies made it possible to recognize several seismogenic
faults in the Apennine active extensional belt (Barchi et al., 1999; Galadini and Galli, 2000; Maschio et al., 2005; Brozzetti,
2011) and, in some cases, to document, through paleo-seismological data, their timing of reactivation during the Holocene
(Galli et al., 2020). Furthermore, in recent years, new technologies have made it possible to reconstruct fault patterns with very
high precision, thus allowing to constrain the fault structures at the surface at sub-meter scale resolution (e.g., Westoby et al.,
2012; Johnson et al., 2014; Cirillo, 2020; Bello et al., 2021b). Accurate geophysical prospections (e.g., ground Penetration
Radar), aimed at the study of historical earthquakes' surface faulting, allowed to investigate the sub-surface (e.g., Gafarov et
al., 2018; Ercoli et al., 2013, 2021), but high-resolution only providing constraints at shallow depths (few tens of m).
Conversely, the geometries of the faults at depth are poorly reliable since deep geological and geophysical constraints are often
lacking.



In fact, in the last decades, seismic reflection prospecting and deep-well exploitation for hydrocarbon research, avoided the
area affected by active extension, and focused on the eastern front of the chain and on the Adriatic-Bradanic foreland basin
system (ViDEPI: www.videpi.com).
The availability of high-resolution seismological datasets, to be integrated with geological ones, provides a new opportunity
to image the tri-dimensional shape of the sources.
Datasets characterized by highly precise re-locations of hypocenters were collected during recent seismic sequences associated
with medium- and high-magnitude earthquakes (Chiaraluce et al., 2004, 2005, 2011, 2017; Totaro et al., 2013, 2015). These
sequences include thousand of earthquakes  in confined rock volumes which appear to roughly connect with the fault traces
mapped at the surface. Therefore, such distributions of earthquakes are generally referred to as ongoing rupture processes
affecting an entire, or wide portion of, seismogenic faults.
In some cases, (Chiaraluce et al., 2017; Valoroso et al., 2017) very high-resolution hypocenter locations, as well as reflection
seismic lines, allow to clearly highlight the seismogenic structures at depth (Lavecchia et al., 2011, 2012, 2015, 2016).
A number of favourable factors make the Calabria-Lucania boundary (southern Apennines) an -interesting study area for the
reconstruction of 3DFM, using good quality seismological data.
This area includes the northern sector of the so-called "Pollino seismic gap" (Fig. 1), in which paleo-earthquakes up to M=7
are documented (Michetti et al, 1997; Cinti et al., 1997, 2002) whereas the location and size of seismogenic sources are a
matter of debate (Michetti et al., 2000; Cinti et al., 2002; Papanikolaou and Roberts, 2007; Brozzetti et al., 2009, 2017a).
Recently, using structural-geological and morpho-structural survey techniques, Brozzetti et al. (2017a) mapped a set of active
faults between the Mercure, Campotenese, and Morano Calabro Quaternary basins (Fig. 1a).
During the 2010-2014 time interval, this area was affected by a low to moderate instrumental seismicity (Pollino seismic
sequence), climaxing at the 25 October 2012, $M_w$ 5.2 Mormanno earthquake, with thousands of recorded events (Totaro et al.,
2013, 2015). During the sequence, two other strongest events occurred close to the village of Morano Calabro: the 28 May
2012 ($M_w$ 4.3), and the 6 June 2014 ($M_w$ 4.0) earthquakes (Fig. 1b).
The whole seismicity was arranged in two major clusters and a minor one (Totaro et al., 2015). Each major cluster was
associated with one strong event and was generated by an independent seismogenic structure (Brozzetti et al., 2017a).
The pre-existence of a seismic network, that was implemented after the start of the sequence, made it possible to increase the
precision of the hypocentral determination and to relocate the events after an accurate selection, providing a high-quality
database (Totaro et al., 2013, 2015; Brozzetti et al., 2017a).
In such context, the main purposes of this work are to:
•    reconstruct the 3DFM activated by the Pollino 2010-2014 seismic sequence;
•    investigate, at depth, the possible interactions between the various seismogenic sources determining the cross-cut

94       relationships between the faults having different strikes, dip-angles, and timing of activation;

•    provide the geometric parameters of the sources aimed at estimating the maximum expected magnitudes based on the

96       defined 3D source.


Finally, we discuss some methodological aspects which are of general interest for researchers approaching three-
dimensional seismotectonics in any structural context. These aspects dwell on the improvements that the proposed
procedure provides to the definition of the source model, the limits of the method imposed by the type and quality of the
available data, and the possible causes of errors with the relative ranges of variation.

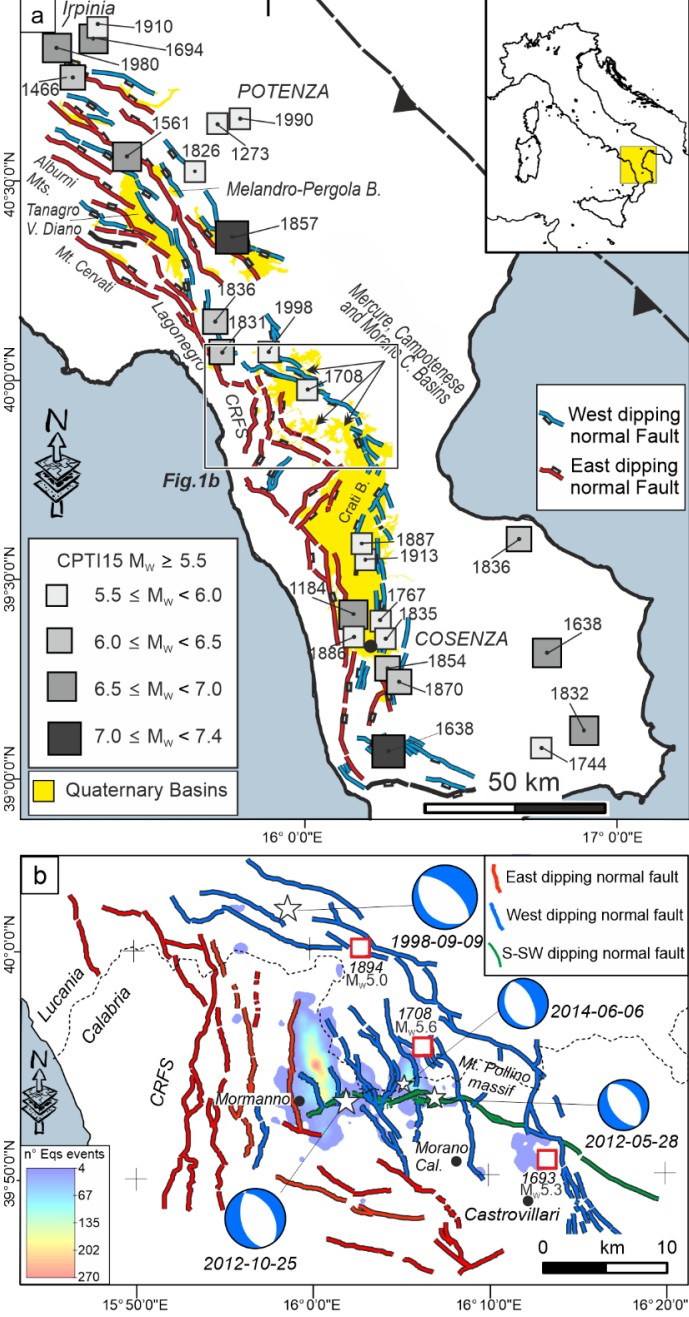




**Figure 1:** Seismotectonic context of the study area. (a) Active faults of the Southern Apennines with major historical and instrumental earthquakes from Parametric Catalogue of Italian Earthquakes, CPTI15 v3.0 (Rovida et al., 2020, 2021). (b) Distribution of the 2010-2014 Pollino seismic sequence focal mechanisms are the events with Mw>4.0 (Totaro et al., 2015, 2016). Normal faults between the Mercure, Campotenese, Morano Calabro and Castrovillari Quaternary basins are reported (after Brozzetti et al., 2017a).

## 2 Seismotectonic framework

### 2.1 Geological Setting

The Mt. Pollino massif is located at the Calabrian-Lucanian boundary (Fig. 1) in a sector of the Apennines structured during the Middle-Late Miocene contractional tectonics that affected the western Adria Plate (D'Argenio, 1992; Patacca and Scandone, 2007; Ietto and Barilaro, 1993; Iannace et al. 2004, 2005, 2007).

The surface geology in this area is characterized by the superposition of two main tectonic units each derives from different paleogeographic domains. These are represented, from bottom to top, by:

– the "Apenninic" units (or "Panormide"), characterized by carbonate platform, including the Verbicaro and Pollino Units with an age from Triassic to Early Miocene, locally intruded by basaltic rocks (Ogniben, 1969, 1973; Amodio-Morelli et al., 1976; Iannace et al., 2007; Patacca and Scandone, 2007; Vezzani et al., 2010; Tangari et al 2018);

– the "Ligurian" units, that consist of ophiolites and deep-sea sedimentary deposits derived from the Western Tethys oceanic basin (Ogniben, 1969, 1973; Amodio-Morelli et al., 1976; Liberi et al., 2006; Liberi and Piluso, 2009; Filice et al., 2015).

During uppermost Miocene-Pliocene times, the folds and thrusts edifice was displaced by WNW-ESE-striking left-lateral wrench faults, among which, the Pollino fault (POL) played an important role (Fig. 1b) (Grandjacquet, 1962; Ghisetti and Vezzani; 1982; Van Dijk et al., 2000).

Subsequently, regional-scale extensional faults systems, consisting of E- and W-dipping conjugate normal faults, dissected the Tyrrhenian side and the core of the orogen which assumed a typical basin and range relief.

The Quaternary extensional phase also caused the reactivation of the previous strike-slip structures. In particular, the reactivation of the POL, with normal to normal-oblique kinematics, has been documented at least since the Early-Middle Pleistocene (Ghisetti and Vezzani, 1982, 1983, Brozzetti et al., 2017a).

Actually, the age of onset of the extensional tectonic is still under discussion; it is referred by some authors to the Early Pleistocene (Ghisetti and Vezzani, 1982; Schiattarella et al., 1994; Papanikolaou and Roberts 2007; Barchi et al., 2007; Amicucci et al., 2008; Brozzetti, 2011; Robustelli et al., 2014), whereas it would not be older than the Middle Pleistocene, according to others (Caiazzo et al., 1992; Cinque et al. 1993; Hypolite et al., 1995; Cello et al., 2003; Giano et al., 2003; Spina et al., 2009; Filice and Seeber, 2019).

In the Campania-Lucania and north-Calabria sectors of the southern Apennines, the active extensional belt includes three main





alignments of normal faults and Quaternary basins, arranged in a right-lateral en-echelon setting (Fig. 1a). From north to south
they are: i) northern alignment: including the Irpinia fault and the Melandro-Pergola and Agri basins; ii) intermediate
alignment: developing from the Tanagro-Vallo di Diano basins to the Mercure-Campotenese and Morano Calabro basins, and
iii) southern alignment: from the Castrovillari fault to the southern Crati basin (Pantosti and Valensise, 1990, 1993; Ascione
et al., 2013; Galli and Peronace, 2014; Ghisetti and Vezzani, 1982, 1983; Barchi et al., 1999, 2007; Blumetti et al., 2002;
Amicucci et al., 2008; Maschio et al., 2005; Villani and Pierdominici, 2010; Brozzetti, 2011, Faure Walker et al., 2012;
Brozzetti et al., 2009, 2017a, 2017b; Robustelli et al., 2014; Sgambato et al., 2020; Bello et al., 2021a).
All along the above alignments, the geometry and kinematics of the major normal faults are kinematically compatible with a
SW-NE direction of extension (Maschio et al. 2005; Brozzetti, 2011; Brozzetti et al., 2009; 2017a). A similar orientation of
the T-Axis is obtained from the focal mechanisms of the major earthquakes from CMT and TDMT databases (Pondrelli et al.,
2006; Scognamiglio et al., 2006; Montone et al., 2012; Totaro et al., 2016) and from GPS data (D'Agostino et al., 2014).
Cheloni et al. (2017) demonstrated, from geodetic GPS and DInSAR analysis, that the Pollino area was affected by important
deformation rates during the 2010-2014 seismic activity, with increasing and decreasing of slip values due to the temporal and
spatial behavior of the recorded seismicity (Passarelli et al. 2015). The present activity of these normal faults systems is firstly
suggested by the control exerted on the distribution of seismicity, as shown by the location of upper crustal instrumental
earthquakes (ISIDe database, Working Group INGV; Brozzetti et al., 2009; Totaro et al., 2014, 2015; Cheloni et al., 2017;
Napolitano et al., 2020, 2021; Pastori et al., 2021) and of destructive historical events (Fig. 1, Rovida et al., 2021).
The area affected by the 2010-2014 seismicity extends from the Mercure basin to the Campotenese and Morano Calabro basins
that are along the intermediate extensional fault-alignment described above. In this area, recent structural-geological works
highlighted three main sets of genetically-linked normal and normal-oblique active faults (Brozzetti et al., 2017a; Figs 1b, 2).
The first one, E- to NNE-dipping, referred to as the Coastal Range Fault Set (CRFS; red lines in Figs 1b, 2) encompasses four
sub-parallel major fault segments which, from west to east are: Gada-Ciagola (GCG), Papasidero (PPS), Avena (AVN) and
Battendiero (BAT) faults. Their strike-direction varies southward from N-S to WNW-ESE.
The other two sets strike ~NW-SE, and dip ~SW (blue lines in Figs 1b, 2). The western one, developing from Rotonda to
Campotenese villages through the 2010-2014 seismic sequence epicentral area, consists of two main right-stepping en-echelon
fault segments. They are referred to as ROCS system, and include the Rotonda-Sambucoso (RSB) and Fosso della Valle-
Campotenese (VCT; Fig.1b). The eastern set, including the en-echelon Castello Seluci-Piana Perretti-Timpa della Manca
(CPST), the Viggianello-Piano di Pollino (VPP) and the Castrovillari (CAS) faults, represents the break-away zone of the
Quaternary extensional belt. In the area between these two W-dipping sets, the W to NW-dipping Morano Calabro-Piano di
Ruggio (MPR) and Gaudolino (GDN) faults, show evidences of Late Quaternary activity (Brozzetti et al., 2017a; Fig. 1b).







**2.2 Earthquake/fault association and kinematics of the 2010-2014 seismic sequence**

In the study area, the POL and the adjacent CAS faults (Fig. 1b) are the most studied structures from the seismotectonic point
of view, as, based on the results of paleoseismological investigations (Michetti et al., 1997, 2000; Cinti et al., 1997, 2002),
they are considered active and capable of producing strong earthquakes and surface faulting. In fact, according to the
aforementioned literature, both the faults were associated with at least two strong earthquakes, (M 6.5 and M 7.0), occurred in
2000-410 B.C. and 500-900 A.D.
The epicenter of the 8 January 1693 earthquake (Fig. 1b) is also located within the hanging wall block of the CAS and in the
footwall block of the MPR fault, some km eastward of the 2012 and 2014 Morano Calabro strong events, then leading to
exclude the MPR as the causative fault.
Nevertheless, this event, which was first reported in the CFTI5Med Catalogue (Guidoboni et al., 2018, 2019) with $M_w$ 5.7,
was recently reduced to $M_w$ 5.2 (Tertulliani and Cucci, 2014).
The epicenter locations of the $M_w$ 5.5, 1708 (Rovida et al., 2021) and $M_w$ 5.1, 1894 earthquakes, close to the northern
termination of the RSB and within its hanging wall, allows hypothesizing the latter fault as the possible seismogenic source.
For what concerns the instrumental seismicity, the main event recorded in the Pollino area is the $M_w$ 5.6 Mercure earthquake
(9 September 1998, Fig. 1b), which was followed by some hundred aftershocks. Despite a preliminary attribution to the
Castelluccio fault (Michetti et al., 2000), this earthquake was associated by Brozzetti et al. (2009) with the SW-dipping CPST
(Fig. 1b), located some km to the NE of the Mercure basin.
The 2010-2014 Pollino seismic sequence was triggered by extensional upper crustal deformations, as highlighted by the focal
mechanisms of the three strongest earthquakes ($M_w$ 5.2, 25 October 2012-Mormanno, $M_w$ 4.3, 28 May 2012-Morano Calabro
and $M_w$ 4.0, 6 June 2014-Morano Calabro). All the associated WSW-ENE oriented T-axes are also ~parallel to the geological
and seismological minimum compression axis provided by the tensorial analysis in the neighboring Mercure area (Brozzetti
et al., 2009; Ferranti et al., 2017), or derived from borehole breakout investigations (Montone et al., 2004; Pondrelli et al.,
2006), and GPS data (D'Agostino et al., 2014).
The available focal solutions of the Pollino 2010-2014 seismic sequence, display W-dipping seismogenic planes which well
correlate with the Quaternary normal fault segments recognized in the epicentral area (see sect. 2.1). The coherence between
field and seismological data is even more evident using the dataset provided by Totaro et al. (2015, 2016) and Brozzetti et al.
(2017a) which suggest N-S to NNW-SSE-striking, W-dipping, seismogenic sources.
Correlating the hypocenters distribution at depth with the active faults highlighted at the surface, the seismogenic source of
the 25 October 2012 Mormanno Earthquake ($M_w$ 5.2), also responsible for the westernmost and larger seismicity cluster (Fig.
1b), is identifiable in both the segments of the WSW-dipping ROCS system (RSB and VCT). These faults dip at surface 70°-
75° and would reach, at depth, a dip of ~55° (Brozzetti et al., 2017a).



Through similar reasonings, the WSW-dipping MPR fault was suggested to be the causative fault of the eastern, Morano
Calabro, cluster (Fig. 1b) and of its two major events ($M_w$ 4.3, 28 May 2012 and $M_w$ 4.0, 6 June 2014). This fault extends for
~7 km in the N170 direction and is co-axial with the W-dipping nodal planes of the two main events of the sequence (Fig. 1b).
The partial reactivation of the CAS could be invoked to explain the minor seismicity cluster recorded at the eastern side of the
study area, although some of the events seem to be located in its footwall.

**3 Material and Methods**

**3.1 Structural survey and fault kinematic analysis**

A series of fieldwork campaigns between 2018 and 2020, were performed to collect fault-slip data in the study and
neighbouring areas, at 1:25.000 scale. These measurements were used to integrate the geological-structural data and to
constrain the Quaternary extensional faults provided in Brozzetti et al. (2017a). In field, we initially used traditional structural
analysis techniques. We integrated the data with data collected through digital mapping, by using the Fieldmove app/software
(PetEx Ltd., version 2019.1) installed on a tablet computer. All these data (shown in Fig. 2) were managed in a GIS database
elaborated through ArcGIS v.10.8 (ArcMap©). Fig. 2 also shows the location of the kinematic survey sites that are structurally
homogeneous outcrops or groups of adjoining outcrops that fall within a maximum distance of 500 m, that is within the
diameter of each small circle on the scheme (more detailed localizations in Supplementary Fig. 2).
The overall fault-slip data set was first subdivided in minor and local homogenous kinematic sub-sets represented as pseudo-
focal mechanisms using FaultKin 8 software (Marrett and Allmendinger, 1990; Allmendinger et al., 2012) (Fig 3). The
obtained beachballs show the computed pseudo-focal mechanism associated with a couple of average fault plane/average slip
vector (A.f.p. and A.s.v. respectively, in Fig. 3) obtained through Bingham statistic analysis from the fault population collected
in each survey site.
The fault/slip data were subsequently inverted in order to reconstruct the long-term (geological) stress field to be compared
with the seismological tensor obtained from available focal mechanisms (see following sec. 3.2). Such comparison allowed
verifying the persistence of the stress field over time, at least from the Middle Pleistocene to Holocene times.

**3.2 Geological and seismological stress tensor inversion**

To investigate the coherence between the geological and the present-day (seismological) stress fields, we applied stress tensor
inversions to the available fault-slip data (Fig. 2, 3) and focal mechanisms (Fig. 4).
We used the inversion procedure proposed in Delvaux and Sperner (2003) (Win-Tensor software) and applied it, separately,
on the different datasets. The procedure attempts to compute the orientation of the three principal axes of the stress ellipsoid
($\sigma_1$, $\sigma_2$, $\sigma_3$) and the stress ratio $\Phi = (\sigma_2-\sigma_3)/(\sigma_1-\sigma_3)$ that optimize the misfit Function (i.e., F5). The latter is built to i) minimize



the slip deviation between the observed slip line and resolved shear stress (30° misfit value is not expected to be exceeded),
and ii) favor higher shear stress magnitudes and lower normal stress to promote slip on the plane.
The inversion procedure provides for the preliminary (kinematic) analysis of data using an improved version of the Right
Dihedron method (Angelier and Mechler, 1977) to determine the starting model parameters (e.g., the reduced stress tensor).
The stress ellipsoid is then computed through a 4D grid-search inversion involving several runs during which the reduced
tensor is rotated around each stress axis, with a decreasing range of variability (from ±45° to ±5° and the full range of Φ values
is checked [0-1]). Each step attempts to find the parameters that minimize misfit function and that are used as a starting point
for the next run (see for details Delvaux and Sperner, 2003).
The geological data input consists of 268 quality selected fault/slip data measured along the fault systems of the study area
(Fig. 2, 3). During the formal inversion, the same weight value was assigned to each fault giving the same quality factor
assigned to the slickenlines.
The seismological data input is represented (initially) by both nodal planes of each focal mechanism; afterward, the plane that
is best explained by the stress tensor in terms of the smallest misfit function is considered as the actual fault plane (Delvaux
and Barth, 2010).
The inverted seismological data are represented by focal mechanisms ($2.7 \leq M_w \leq 5.0$) taken from Totaro et al. (2015, 2016) and
reported in Fig. 4. An exponential weighting factor (corresponding to the earthquake magnitudes) has been assigned to account
for the prevailing kinematics of the most energetic events.
The final inversion includes only the fault- and focal-planes that are best fitted by a uniform stress field (Gephart and Forsyth,
1984). The stress inversion results are shown in Fig. 5.







**Figure 2:** Structural Map at the Calabrian-Lucanian boundary (after Brozzetti et al., 2017a) with location of fault-slip data measurements. Fault Key: CRFS= Coastal Range Fault Set; GCG= Gada-Ciagola fault; PPS= Papasidero fault; AVN= Avena fault; BAT= Battendiero fault; ROCS= Rotonda-Campotenese fault system; VCT= Fosso della Valle-Campotenese fault; RSB= Rotonda-Sambucoso; CVN= Cozzo Vardo-Cozzo Nisco fault; MPR= Morano Calabro-Piano di Ruggio fault; VPP= Viggianello - Piano di Pollino fault; GDN= Gaudolino fault; POL= Pollino fault; CAS= Castrovillari fault; SDD= Serra Dolcedorme fault; PAC= Monte Palanuda – Campolungo fault; CPST= Castello Seluci-Piana Perretti fault.





**Figure 3:** Kinematic analysis and pseudo-focal mechanisms obtained from fault/slip data using the software FaultKin 8 (Allmendinger et al., 2012). Pseudo-focal mechanisms are boxed with different colors on the basis of the fault system to which they belong (color key as in the map of Figure 1, 2). For each fault system the density contour of the T-axis computed for each focal mechanism is reported (lower hemisphere projection). A.s.v.=Average striae value, A.f.p.=Average fault plane. Numbers


in the rectangles (top left of each focal mechanism) refer to the group of fault/slip data belonging to or neighbouring of a single
site (location in Figure Supplementary 2).

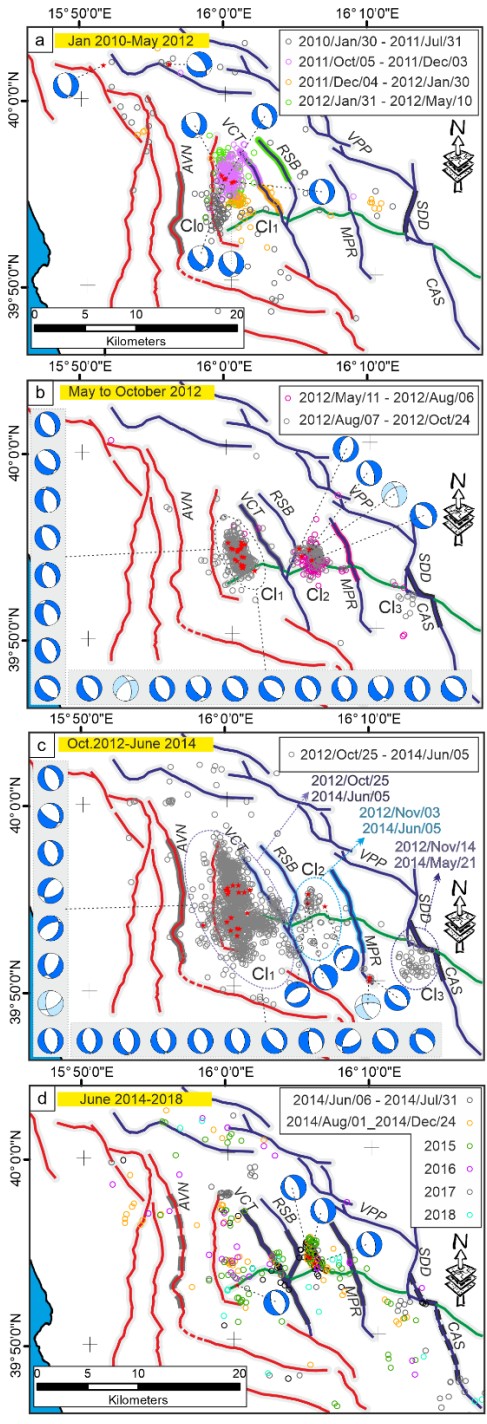



**Figure 4:** Time-space evolution of the 2010-2018 seismic activity in the Pollino area. Each panel shows the distribution of focal mechanisms and epicenters concentrated in a series of neighbouring clusters numbered as Cluster 0, 1, 2, and 3 (Cl0, Cl1, Cl2, Cl3) from west to east according to their activation time. See the main text (section 4.2) for the sequence description.

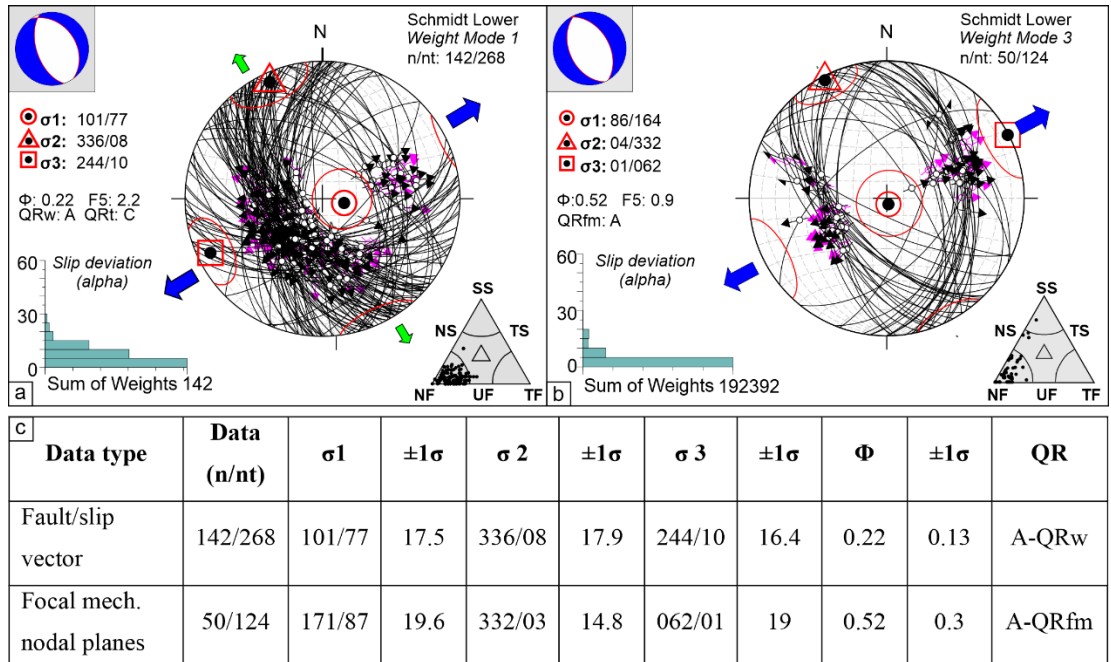

| c | Data type | Data (n/nt) | σ1 | ±1σ | σ 2 | ±1σ | σ 3 | ±1σ | Φ | ±1σ | QR |
|---|-----------|-------------|-----|------|------|------|------|------|------|------|--------|
| | Fault/slip vector | 142/268 | 101/77 | 17.5 | 336/08 | 17.9 | 244/10 | 16.4 | 0.22 | 0.13 | A-QRw |
| | Focal mech. nodal planes | 50/124 | 171/87 | 19.6 | 332/03 | 14.8 | 062/01 | 19 | 0.52 | 0.3 | A-QRfm |

**Figure 5:** Stress inversion results for the geological- (a) and seismological (b) data. On the lower hemisphere Schmidt nets, the pairs fault plane/slickenline (a) and focal plane/kinematic indicators (rake) (b) are reported (great circles represent the fault planes; the dark and pink arrows indicate the measured slip directions (or rake) and resolved shears, respectively). The histograms represent the corresponding misfit angles vs. the number of data points; nt = total number of fault data; n = number of successfully inverted fault data; σ1, σ2, σ3 = principal stress axes; Φ = stress ratio = (σ2-σ3)/(σ1-σ3); the quality ranking factors (QR) and the stress inversion parameters with associated uncertainties (1σ standard deviations) are listed in panel (c). On the small upper left nets, the computed stress field represented as a focal mechanism is reported. The triangles reported on the lower right corners of each panel (a) and (b) show the kinematic classification of data according to Frohlich (2001). (c) Geological and seismological stress tensor parameters computed starting from slip-vector measurements collected along the investigated fault systems (Figs. 2, 3) and 2.7≤Mw≤5.0 focal mechanisms (see. Sect. 3.3 and Fig. 4), respectively. Key: nt = total number of data (e.g., plane/slickenline); n = inverted data; σ1, σ2, σ3 = principal stress axes; Φ = stress ratio = (σ2-σ3)/(σ1-σ3). QR = quality ranking: AQRw as in Sperner et al. (2003) and A-QRfm as in Heidbach et al. (2010).





### 3.3 Hypocenter location

To better characterize the 3D features of the structures lying in the region of the Mercure-Pollino sequence and to frame it in the geological scenario of the Calabrian-Lucanian border, we performed a high-quality hypocenter location. In previous works by Totaro et al. (2013 and 2015) and Brozzetti et al. (2017a), focused on the Mercure-Pollino sequence, the seismicity occurred in the time period 2010-2014 was analyzed. This allowed the authors to well characterize the seismic activity and to provide preliminary interpretations for geological features of the study area.

In this study, we sensibly enlarged the time window for earthquake analyses, with the data for all the earthquakes that occurred in the area between January 2010 and October 2018 (local magnitude greater than 1.0 and hypocentral depth range 0-30 km) collected from the Istituto Nazionale di Geofisica e Vulcanologia (INGV) Bulletin and the University of Calabria database. Automatic and manually revised P- and S-wave arrival time picks have been selected for this dataset. The recording network, including both temporary and permanent stations managed by the University of Calabria and INGV (D'Alessandro et al., 2013; Margheriti et al., 2013), consisted of 61 stations with a maximum epicentral distance of 150 km (Supplementary Fig. 1). We computed accurate absolute hypocenter locations by applying first the non-linear Bayloc earthquake location algorithm (Presti et al., 2004, 2008) and subsequently the double-difference relative location method HypoDD (v.2; Waldhauser, 2001), and using the 3D P-wave velocity model by Orecchio et al. (2011). The Bayloc algorithm gives for each earthquake a probability density cloud with shape and size related to the main factors involved in the location process (e.g., network geometry, picking errors), and allows a generally more accurate estimate of hypocenter parameters and location uncertainties with respect to the more commonly used linearized location methods (see e.g., Lomax et al., 2000; Husen and Smith, 2004; Presti et al., 2008). The application of the Bayloc algorithm to the collected dataset provided, on average, horizontal and vertical errors of the order of 1.0 and 1.5 km, respectively, and allowed us to obtain a well-constrained database that has been used as starting point for the subsequent analyses. As the second step, we applied the HypoDD algorithm, which minimizes phase delay-time residuals between pairs of events recorded at common stations (Waldhauser and Ellsworth, 2000). We computed the delay times from each event to its 30 nearest neighbors within 10 km distance, and to further ensure the robustness of the double-difference inversion. Only event pairs with at least eight phases observed at common stations were used. The final relocated dataset consists of 3109 events (Fig. 4 and Supplementary Fig. 1).

Before the 2010-2014 Pollino sequence, the instrumental data available within a range of nearly 75 km from the Mercure basin, referred to background seismic activity (Frepoli et al., 2005; Castello et al., 2006; Brozzetti et al, 2009). In this framework, the only phases of significant seismic activity which affected the region, were the above mentioned 2010-2014 Pollino sequence (Fig. 4 and Supplementary Fig. 1) and the moderate magnitude 1998-1999 Mercure seismic sequence that developed in the northern part of the homonym Quaternary basin (Guerra et al., 2005; Arrigo et al., 2005; Brozzetti et al., 2009). It started on September 1998 and lasted several months, showing some similarities to the recent Mercure-Pollino sequence (e.g., prevalent kinematics of focal mechanisms and hypocentral depth range). We explored the data available for this seismic sequence in order to compute a high-quality earthquake location, following the procedure already described for the 2010-2018 seismic

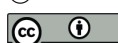



activity. Since the recording network operating during the 1998-1999 seismic phase was significantly different from today in
terms of number of stations deployed in the region and their spatial distribution, the available data do not allow to reach the
high level of constrain needed to perform the 3D structural model reconstruction.


**3.4 3D Model building**

The 3D Fault Model (3DFM) of the potentially seismogenic structures involved in the Pollino earthquakes was performed
integrating the detailed Quaternary fault pattern provided in Brozzetti et al. (2009, 2017a) and integrated with new
constrains from this paper, with the high-quality 2010-2018 seismicity dataset.
We applied the methodology defined by the Community Fault Model of Southern California (Nicholson et al., 2014; Nicholson
et al., 2015; Plesch et al., 2014) which was also used, in previous works, to depict the subsurface geometry of the faults
reactivated during recent Italian earthquakes (Lavecchia et al., 2017; Castaldo et al., 2018; Bello et al., 2021a).
In the present work, the uncertainties in identifying the active fault segments and determining the fault-earthquake
associations were overcome thanks to the recent work of Brozzetti et al. (2017a) who discussed these aspects but without
investigating the subsurface geometries of the sources.
We obtained the latter by interpreting as seismogenic fault-zone, the well-confined deformation volumes illuminated by the
clustering of the hypocenters and using, for the 3D reconstruction, the Move suite software v. 2019.1 (Petroleum Experts Ltd). We
describe in detail the steps of the 3D model building of the seismogenic sources in the following sections.

**4 Results**
**4.1 Geological and Seismological Stress Tensors**

The computed geological stress tensor (Fig. 5) shows a relevant percentage of fault/slip vector pairs (~53%) consistent with a
uniform extensional stress field which is characterized by a N244 trending- and sub-horizontal σ3. The stress ratio
Φ=0.22±0.13 and the rank quality is QRw=A (ranking as in Sperner et al., 2003). Nearly all the kinematic axes related to the
inverted data belong to a normal-fault regime according to the triangle classification in Frohlich (2001) (see Fig. 5a).
It is worth noticing as 76% of the successfully inverted fault/slip vector pairs are related to the active fault planes belonging to
the E- and W-dipping domains (Fig. 5a) while the remaining 24% include data related to the S-dipping system (CVN and
POL). The latter evidence is consistent with the prevalent activation in the Late Quaternary of the E- and W-dipping fault
systems.
The seismological stress tensor (Fig. 5b) obtained from inverting 50 actual fault planes (nt = 124 nodal planes), shows a normal
fault regime with an ENE-WSW trending and sub-horizontal σ3 (N062/01 ±19). The stress ratio Φ=0.52 ±0.3 and the rank



quality is QRfm=A (ranking as in Heidbach et al., 2010). Most of the nodal planes show normal-fault kinematics (see the
triangle diagram on the lower right corner of Fig. 5b).
Finally, in both the inversions, a normal-fault regime with sub-horizontal and collinear (~SW-NE trending) σ3-axis has been
obtained. This result confirms the coherence between the geological and the present-day stress field and the persistence of this
extensional regime since the Middle Pleistocene.

**4.2 Time-space evolution of the Pollino sequence**

The 2010-2018 seismic activity interval in the Pollino-Mercure area followed a peculiar evolution over time schematized in
Fig. 3a-d. The distribution of epicenters concentrated in a series of neighboring clusters which were numbered as Cluster 0, 1,
2, and 3, from west to east, also according to their activation time. Such clusters, independent and unconnected, to each other,
can be related to fault segments that are not in an along-strike continuity.
Cluster 0 includes the earliest (30/01/2010 - 31/07/2011), low magnitude ($1.0 \leq M_L \leq 2.9$) activity located in an NNE-SSW
lengthened sector at the western boundary of the epicentral area. It is delimited westward by the more external segment of the
E-dipping CRFS. In terms of generated events, this cluster was rather intermittent with periods characterized by moderate
activity (Figs. 4a,c) alternating to substantially inactive ones (Figs. 4b,d). No significant seismicity occurred here during the
2015-2018 time span.
Cluster 1 was active after 05/10/2011 and during the entire 2010-2014 seismic sequence. It extended continuously, either
northward and southward, reaching an NW-SE length of ~12 km (Fig. 4a-c). It comprehends the higher number of earthquakes
and is largely the major cluster as regards the wideness (~60 km$^2$) and energy release. It includes 30 events with $M_L \geq 3.0$
besides the 25 October 2012 strongest event of the whole Pollino seismic activity. During the 2015-2018 interval, Cluster 1
area was affected by low seismic activity, mostly distributed in its northern and southern portions; conversely, its central part,
where epicenters were particularly dense between 2011 and 2014, became nearly silent.
Overall, the surface extent of Cluster 1, which partly overlaps with Cluster 0, is limited eastward by the W-dipping RSB and
VCT faults. Its southern boundary nearly coincides with the southeastern continuation of the AVN fault (PAC, Fig. 4c).
Cluster 2 was activated in May 2012 in the sector between the two WSW-dipping RSB and the MPR faults. It elongates in N-
S direction, for ~7 km to the northwest of the Morano Calabro town. Afterwards, it was nearly continuously active, particularly
during the periods May-October 2012 and June-October 2014 (Fig. 4b,c); also in the period 2015-2018, significant seismicity
persisted (Fig. 4d). Cluster 2 includes mainly low-magnitude events besides the strongest ones of 28 May 2012 and 6 June
2014 and three other earthquakes with $3.0 \leq M_L \leq 3.5$.
Further east, in the sector comprised between MPR and the alignment VPP-SDD-CAS faults, a minor seismicity cluster
(Cluster 3) develop since December 2011 (Fig. 4a). Since then (2011-2018) it was affected by poor and low-magnitude
seismicity, which however was clearly above the threshold of background seismicity, with two $M_L$=3.0 events (Fig. 4a-d).



### 4.3 3D Fault Model building

Following the approach adopted and described in literature for the 3D fault model building (Lavecchia et al., 2017; Castaldo et al., 2018; Bello et al., 2021a), we created several sets of closely spaced transects (half-width=2 km) to cross and sample the seismogenic fault zones in different directions (Fig. 6). The first two sets (oriented SW-NE and NW-SE) are respectively ~orthogonal (e.g., sections 1, 2 in Fig. 6) and ~sub-parallel (e.g., sections 3-6 in Fig. 6) to the ROCS (VCT and RSB), and MPR active fault. A further NNE-SSW-striking set of transects was traced ~orthogonal to the active fault alignment bounding eastward the area affected by the 2010-2014 seismic sequence, which includes the CPST and VPP faults (sections 7 and 8 in Fig. 6).

The 3DFM building  was carried out following three steps graphically depicted in Fig. 6 and synthetically described below.

*Step 1-Extrusion of fault traces to shallow depth*

The traces of the Quaternary faults, belonging to both E- ad W-dipping sets (Fig. 7) are "extruded" (i.e., projected downward three-dimensionally) along-dip to reconstruct the so called "fault ribbons" (Fig. 7a). These latter are extrapolated to a pre-set depth of 2 km b.s.l, according to the dip-angle of the fault planes measured at the surface during fieldwork campaigns both of this work and from previous literature (Brozzetti et al., 2009, 2017a). In the absence of these data, we assumed a fixed $60^{\circ}$ dip-angle.  The obtained ribbons are rimmed upward by the topographic surface represented as a  10 m-resolution DEM (Tarquini et al., 2012). In the model, they close at the fault tip and their relationships with the neighboring ribbons depend on the geometry of the transfer zone between adjoining faults.

*Step 2- Down-dip extrapolation of the faults along seismological sections*:

Starting from the analysis of the seismological transects (Fig. 6), the fault extrapolation at depth is based on the assumption that the seismogenic volumes, (i.e., the main clusters of hypocenters), illuminate the portion of the activated fault zones. In particular, the depth geometries were traced by connecting the surface "ribbons" (step 1) with the zones at depth where seismicity is more dense (i.e., where the concentration of hypocenters is higher; Fig. 7b,c) downward to the base of the seismogenic layer.  The latter, according to the model proposed in Brozzetti et al. (2017a), corresponds to an E-dipping basal detachment.

We also considered the attitudes of the preferential fault planes from focal mechanisms falling within each section. In general, this step is relatively simple in those cases where seismicity originates on fault zones having regular shape, whereas it gets complicated in areas, as the study one, where multiple faults, characterized by sharp strike-changes, interact. In such cases, seismicity distributes in geometrically complex volumes and the hypocentre clusters have irregular contours, possibly because include groups of events generated on adjacent faults.

*Step3- Building of 3D fault surfaces through Delaunay triangulation method*

This method automatically allows reaching the final 3D reconstruction of the seismogenic faults (Fig. 7c,d) by interpolating all the elements derived by the previous two steps.

After joining at depth the fault ribbons with the subsurface sources, we obtained the seismogenic patches by projecting





on each reconstructed fault plane the seismicity cluster as density contour (Fig. 6d). This depiction allows visualizing,
with a good degree of approximation, the portion of the fault which released most of the seismicity during the considered
time interval.
This methodology made it possible to refer with good confidence the hypocentral distribution of the 2010-2018 seismicity
to the causative faults confirming that Cluster 1 and Cluster 2 well-correlate respectively with the SW-dipping ROCS
(RSB+VCT) and MPR faults. The westernmost concentration of hypocentres (i.,e., Cluster 0), given its location, may be
due either to the deeper portion of the VCT fault or to the E-dipping AVN Fault, which act as the lower boundary of the
W-dipping fault set (Fig. 7c,d).









**Figure 6:** Epicentral (map – upper right) and hypocentral distributions (sections – a to i) of the 2010-2018 seismic activity occurred in the Pollino area. In the vertical cross-sections the earthquakes have been reported also as density contours computed using Kernel Density Estimation. The histograms close to each section show the depth distribution of the sequence. The traces of the overall dataset of serial cross-sections analysed in this study are reported in map view as thin grey lines while the bold lines relate to the sections shown in this figure.

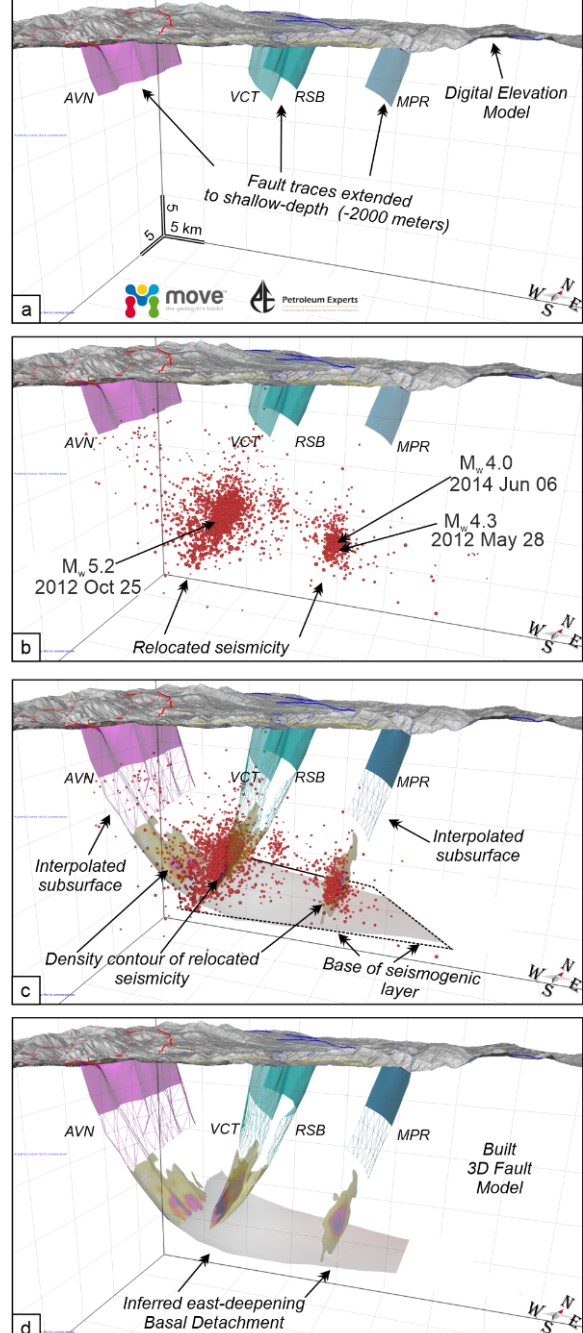



**Figure 7:** 3D fault model building, from surface (represented by a 10 m-resolution DEM – Tarquini et al., 2012) to the base of the seismogenic layer. Faults acronyms as in Figure 2. (a) "Fault ribbons" obtained by extruding the fault traces mapped at surface up to shallow depth (2 km b.s.l.), and considering the fault dip-angles measured in the field. (b) 3D fault model as in (a) with relocated seismicity. (c) Fault extrapolation at (seismogenic) depth through the clusters of hypocenters. The modelled faults connect the ribbons with the zones at depth where concentrations of hypocenters is higher. The density contours of the seismicity and the base of the seismogenic layer are also shown (see also panel d). (d) Final 3D fault model obtained integrating the detailed Quaternary fault pattern with the high-quality 2010-2018 seismicity dataset.

### 4.4 3D Fault Model of the Pollino area Quaternary and active fault system

The 3DFM obtained to the depth of ~10-12 km (Fig. 8), as well as including the seismogenic fault system activated during and after the Pollino seismic sequence (i.e., 2010-2018), represented by CRFS, ROCS, and MPR, also encompasses those faults that, while showing no direct evidence of recent seismic activity, play a significant role in the seismotectonic frame of the area. In other words, we also considered the structures that border the reactivated faults, either beneath or laterally, representing possible barriers to the propagation of coseismic faulting.

We interpreted the westernmost fault structures (i.e., GCG and PPS) whose deep geometry is not strictly constrained by subsurface data, according to the structural extensional style proposed by Brozzetti et al. (2017a). This is coherent with the reconstructions of the active extensional belt of the southern and central Apennines described in the literature (Barchi et al., 2007; Amicucci et al., 2008; Brozzetti et al., 2011, 2017a, 2017b; Lavecchia et al., 2017).

Overall, this style is characterized by an asymmetric extension driven by a low-angle (20° to 35°) E-dipping detachment fault which represents the basal decollement of all the other extensional structures.

In the model, all the faults are traced at the surface with their outcrop dip-angle and evolve downward with nearly-listric geometries to join the detachment at increasing depth from west to east. The latter represents, the structurally controlled base of the seismogenic layer.

We interpreted the GCG (Figs 1b, 8), which crops out at low-angle and overcomes all the other east-dipping faults in terms of slip and associate extension, as the currently inactive break-away zone of such a detachment.

The AVN and BAT (Figs 1b, 8), which are the easternmost E-dipping splays, are suggested to be active and seismogenic, being possibly the causative structures of the Cluster 0 of hypocenters recognized during the 2010-2014 activity (Fig. 4).

In such a model, also the W-SW-dipping ROCS and MPR faults, which we consider the main sources of the 2010-2014 seismic sequence, are downward confined by the E-dipping detachment (Fig. 8a,a1).

Further east, the 3DFM has been widened to include the W-dipping CPST and VPP faults, considered the outer seismogenic front of the extensional system (Brozzetti et al., 2009).

The yellow surface shown in Fig. 8c,d, depicts, in nearly frontal view, the 3D shape of the POL and its westernmost segment (CVN) bounding to the north of the Campotenese basin. The along-strike continuity of this fault is interrupted by the W-





dipping ROCS and MPR faults, coherently with the cross-cut relationships observed in the field (Fig. 1b).
In Fig. 8d, the depth geometry of POL and CVN interrupts the NNE-dipping AVN (violet surface in Fig. 8d) which acts as
the southern and basal boundary of the entire active fault system.
Finally, the comparison between the reconstructed 3D-model and the plotted re-located seismicity evidently shows that almost
the whole 2010-2018 hypocenters correlates with the W-dipping structures but without affecting their southern termination
zones. In other words, no or very few events locate south of the intersection with POL and CVN faults.
This latter observation suggests that although the POL and CVN did not play an active role in the origin of the considered
seismic activity, play a significant role in influencing its distribution.


**Figure 8:** Tri-dimensional model of the extensional system at the Calabrian-Lucanian boundary extrapolated down to the depth of ~10-12 km. The geological-structural map used as a base over a 10 m-resolution DEM is from Brozzetti et al., 2017a. Fault surfaces are those belonging to the seismogenic fault system activated during and after the Pollino seismic sequence (2010-2018), and those playing a key role in the seismotectonic context of the area. The faults belonging to the E-NE-dipping CRFS fault set are represented in red and violet, whereas the antithetic ROCS and MPR faults are shown as blue surfaces (fault acronyms as in Figure 2). Yellow surface is the tri-dimensional surface of the POL and its westernmost segment (CVN) bounding, to the north, the Campotenese basin.



## 5 Discussion

### 5.1 Maximum expected magnitude

Coherently with what is observed in most of the Apennine chain, the upper crustal Pollino seismicity developed in response to WSW- ENE oriented extension. This is well constrained by the focal solutions of the strongest events ($M_w$ 5.2, 25 October 2012; $M_w$ 4.3, 28 May 2012, and $M_w$ 4.0, 6 June 2014 earthquakes) and all the $M_w \geq 3.5$ earthquakes that occurred during the 2010-2014 seismic sequence.

The tensor provided by the inversion of fault/slip data, collected on the Quaternary faults (Figs. 2, 3 and 5a), is also extensional and nearly co-axial to the seismological one. Such consistency suggests that the present stress field is in continuity with the long-term one, which is active at least since the Early-Middle Pleistocene as already suggested by previous works (Papanikolaou and Roberts, 2007; Brozzetti et al. 2009; 2017a).

Comparing the distribution of the whole 2010-2018 seismic activity with the Late Quaternary structures mapped at the surface, we maintain that the ROCS and the MPR faults are respectively suitable as the seismogenic sources for the Mormanno (2012, $M_w$ 5.2) and Morano Calabro (2012, $M_w$ 4.3 and 2014, $M_w$ 4.0) earthquakes.

However, our 3DFM goes beyond establishing the earthquake-structure associations and allows defining a strictly constrained parameterization of the sources and assessing their seismogenic potential.

From the overall fault model of Fig. 9a (the same of Fig. 8 but in zenithal view) we extracted the plan view of the W-dipping seismogenic faults (Fig. 9b). This representation depicts irregularly-shaped seismogenic boxes which are bordered to the east by the fault traces and, on the opposite side, by the projection at the surface of the branch line of each fault from the base of the seismogenic layer. Some of the aforementioned boxes include historical or instrumental earthquakes (colored squares in Fig. 9b) while others are not associated with any significant event.

The performed 3D reconstruction allowed us precisely estimating the effective area extent of all the fault segments, despite their irregular shape and complex segmentation pattern. The calculated areas (Fig. 9c left white column), inserted in the appropriate scaling relationships, provide the maximum expected magnitude that each fault might release in the borderline case in which the coseismic rupture affects its entire plane (Fig. 9c, yellow, blue, and brown columns).

The maximum magnitude values obtained using the regressions as a function of the surface fault length (Fig. 9c, right white column) are listed in Fig. 9c (light-blue, green, and grey columns).

Finally, the two graphs of Figs. 9d (fault area-based scaling relationships) and 9e (fault length-based scaling relationships) allow comparing the values determined, for all the investigated active normal faults, using six different empirical relations (Wells and Coppersmith, 1994; Wesnousky, 2008; Leonard, 2010; Stirling et al., 2013).

It is evident that, for each fault, the maximum expected magnitude reported in Fig. 9d (computed using fault area) is lower than that plotted in Fig. 9e (computed using fault length) and also the range of variation (length of yellow bars on Figs 9d and 9e graphs) is narrower for the values computed on the ground of fault-area regressions.



As the difference in the obtained results is significant, a matter of primary importance concerns which scale relationships are
to be used for the evaluation of the maximum expected magnitudes.
In the light of our results, we suggest that, wherever possible, the selection must be made taking into account the 3D
reconstructed geometry of the faults, which allows determining as precisely as possible its shape and parameters
(Supplementary Table 1).
The extensional faults system of the Calabrian-Lucanian border is markedly asymmetrical (Brozzetti, 2011; Brozzetti et al.,
2017a, 2017b). It is characterized by high-angle W-dipping active faults (ROCS, MPR, VPP, CPST and CAS, Fig. 8a-d)
delimited downward by an E-dipping, low-angle, detachment fault (violet surface in Fig. 8a) from which the high-angle active
splays (AVN and BAT, Figs. 7,8) branch upward.
The 3D reconstruction of the fault system highlights that the depth reached by the W-dipping faults, which strictly influence
their areal extension, depends on their position within the hanging wall of the detachment. In other words, faults with
comparable length at the surface may have significantly different areas, depending on the reached depths.
The CPST, VPP and CAS belong to the easternmost extensional alignment and crop out at great distance from the GCG break-
away zone. Consequently, they intersect at the higher depth of the basal detachment and have the maximum area extent among
the W-dipping fault set (Fig. 9a,d) implying, in turn, different values of the maximum expected magnitude.
From the previous reasoning, it follows that the 3DFM, which allows estimating accurately the subsurface extent and areas of
the seismogenic faults, leads to prefer for the assessment of the associate seismogenic potential, the scaling relationships based
on the fault area.
In our case, by applying this type of regressions to the W-dipping faults identified to be the sources of the 2010-2014 seismic
sequence, we calculated the maximum expected magnitude of $\sim M_w=6.1$ for the VCT and the RSB, and of $\sim M_w= 6.2$ for the
MPR. A value of $\sim M_w=6.4$ could be reached in the case of the complete and concurrent ruptures on both the ROCS segments
(joined RSB+VCT).
It is noticeable that the aforesaid values are sensibly higher than the magnitudes of the earthquakes recorded to date in the
Mercure-Campotenese area (Figs 1b, 9b) suggesting that the considered faults released only partially their seismogenic
potential during historical times.
This inference also agrees with the distribution and evolution of the 2010-2018 seismic activity. The clusters of the relocated
hypocenters concentrated in the deepest parts of the ROCS and MPR faults (Fig. 6) confirming that only a portion of such
faults ruptured during the sequence, without the rupture reaching the surface.










**Figure 9:** (a) Seismotectonic 3D Fault Model in map view. (b) Box representation of the 3D seismotectonic model of the West-dipping seismogenic faults in its detailed segmentation pattern. The associated historical earthquakes from CPTI15 v3.0 ($4.5 < M_w < 6.0$; Rovida et al., 2020, 2021) and the epicentral distribution of the 2010-2018 seismic activity occurred in the Pollino area ($1.0 < M_w < 5.2$) are also reported. (c) Maximum expected magnitude according to scaling laws (Wells & Coppersmith 1994, Wesnousky 2008, Leonard 2010, Stirling et al. 2013) and calculated based on area (A) and length (L) of each fault. (d) Graph showing fault- area based scaling relationships. (e) Fault length-based scaling relationships comparing the values determined for all the investigated active normal faults and using six different empirical relations reported in panel (c).

## 5.2 Seismogenic patches activated during 2010-2014 seismic sequence

An attempt to reconstruct the seismogenic patches activated on the ROCS and MPR faults during 2010-2014 seismic sequence, is shown in Fig. 10. The patches can be considered the reasonable approximation of the actual portion of the faults which broke during the mainshock and the sequence of the early aftershocks.

Operationally, we obtained the boundaries of the patches by projecting the relocated hypocenters on the reconstructed fault surface and depicting their distribution using the Kernel density geostatistical analyst, available as a tool of the ESRI ArcGIS software package.

At the depth at which the hypocenters of the 2010-2014 seismicity concentrate, the two segments of the W-dipping ROCS fault set, can be considered joined to form a single structure, thus a unique seismogenic patch was reconstructed.

The temporal analysis of the sequence, showed that their overall extent was already well defined within the first 72 hours after the major events. Anyhow, inside the surrounding volumes, some seismicity had started before the mainshock and also continued to persist constantly throughout the development of the entire sequence so that they include a percentage $\geq$ of 70% of the whole hypocenters locations.

The delimitation of each seismogenic patch and its subsequent parameterization allowed us to verify if there is a direct correlation between its dimensions and the magnitude released by each fault during the mainshocks.

The patches obtained over the VCT and MPR fault surfaces correspond to the violet contoured area shown in Fig. 10. Their along-strike elongation and area extent can be assumed respectively as the effective Subsurface Rupture Length and Rupture Area (RLD and RA in Fig. 10b, and 10c, respectively, according to Wells and Coppersmith, 1994) associated with the $M_w$ 5.2 Mormanno (on VCT fault) and $M_w$ 4.0 and 4.3 Morano Calabro (on MPR fault) earthquakes.

The parameters obtained for the VCT fault are RLD= 4.9 km and RA= 8.3 km$^2$. The values of RLD= 1.2 km and RA= 3.6 km$^2$ are assessed for the MPR fault.

Introducing the aforesaid parameters in the appropriate scale relationships (Fig. 10b,c) we observe a good agreement, or a slight overestimation, between the theoretical magnitudes based on the Subsurface Rupture Length and the magnitudes of the mainshocks. The values obtained for the VCT fault (causative of the $M_w$ 5.2 Mormanno earthquake) are = $M_w$ 5.3 whereas for the MPR fault (causative of the $M_w$ 4.0 and 4.3 Morano Calabro earthquakes) is $M_w$=4.5.





The magnitude calculated using the RA-based relationships provides values slightly lower than expected for the VCT
(4.9<$M_w$<5.0) and little higher for the MPR (4.5<$M_w$<4.6).
In both cases, however, the magnitude values obtained using the scale relationships differ from those observed by an amount
<0.3.


**Figure 10:** (a) Seismogenic patches activated during the 2010-2014 seismic sequence on VCT and MPR faults. The along-
strike elongation and area extent, shown by black arrows, are assumed to be the effective subsurface rupture length and rupture
area (RLD and RA, according to Wells and Coppersmith, 1994) associated with the $M_w$ 5.2 Mormanno (on VCT fault) and
$M_w$ 4.0 and 4.3 Morano Calabro (on MPR fault) earthquakes, respectively. (b) and (c) show the RLD and RA, respectively,
obtained for both the VCT and MPR faults. A comparison between the theoretical magnitudes (obtainable with scale
relationships) based on the subsurface rupture length and the magnitudes of the mainshocks are also shown.





**5.3 Possible geometric restraints to coseismic rupture propagation**

The high-precision seismological dataset we used, demonstrates that the two main clusters of earthquakes of the 2010-2018 seismicity were generated by as many independent sources related to the sub-parallel, 10 to 15 km-long, ROCS and MPR faults.

Brozzetti et al. (2017a) highlighted that the above seismogenic style, characterized by a perpendicular-to-fault strike evolution of the seismic activity, is unlike from those which followed the major instrumental earthquakes recorded in the Apennine Extensional Belt of Italy in recent years, such as the Colfiorito 1997 ($M_w$ 6.0), L'Aquila 2009 ($M_w$ 6.3) and Norcia 2016 ($M_w$ 6.5) events (Chiaraluce et al. 2011, 2017; Lavecchia e al., 2011, 2012, 2016). They also speculated that this peculiar behavior of the 2010-2014 Pollino seismic sequence could have been controlled by the geometric fault pattern of the area, which is characterized by WSW-dipping seismogenic faults bounded southward by nearly E-W pre-existing structures. These latter which are genetically related to the regional-scale, long-lived, "Pollino lineament" *s.l.*" (Bousquet, 1969, 1971; Ghisetti and Vezzani, 1982, 1983; Knott and Turco, 1991; Van Dijk et al., 2000) determine the abrupt contact between the Apennine carbonate platform unit and the San Donato metamorphic core complex (Grandjaquet 1962; Servizio Geologico Nazionale, 1970; Amodio Morelli 1976). The cross-cut relationships detected in the field between the ROCS-MPR set and POL-CVN, highlighted in our 3D model (Fig. 8d), lead us to exclude the latter fault to have a present seismogenic role, as also supported by the distribution of the instrumental earthquakes which clusterized along with N-S-striking crustal volumes.

However, it cannot be excluded that this significant geological boundary exerts an influence on the southward propagation of the presently active seismogenic faults, driving the eastward transfer of the active extensional deformation belt. This inference is confirmed by the spatial distribution of the hypocentres of the whole 2010-2018 relocated seismicity which, with sporadic exceptions, is confined within the CVN footwall (Fig. 8d).

**6 Conclusions**

We reconstructed in detail the 3D geometry and kinematics of the interconnected fault pattern responsible for the moderate-magnitude earthquakes which recently affected the Pollino area.

The main original outcomes that we have achieved are summarized as follows:

- We computed the geological and seismological stress tensor and demonstrate that they are consistent with a uniform normal faulting regime characterized by an ENE-WSW trending, sub-horizontal σ3. This result confirms the coherence between the long-term and the present-day stress field and the persistence of this extensional regime at least since the Middle Pleistocene.

- The 2010-2018 seismic activity which affected the study area followed a peculiar evolution over time characterized by concentration of epicenters in a series of sub-parallel ~NNW-SSE elongated clusters, independent and unconnected, which





can be related to two major near coaxial WSW-dipping faults possibly splaying from a common east-dipping basal detachment
and almost concurrently releasing seismicity.

-The accurate hypocenter re-locations, obtained by applying the non-linear Bayloc earthquake location algorithm, followed by
the double-difference relative location method HypoDD, and using a 3D P-wave velocity model, provided a high-resolution
seismological dataset. The latter was found to be of excellent quality for the purposes of 3D modelling. The correlation between
the geometry of the active faults at the surface, and the distribution of seismicity at depth, allowed us to reconstruct the 3D
geometry of the seismogenic sources which released the 2010-2018 seismicity. This reconstruction was the interpretative key
to obtain the overall model of the Quaternary and active fault system of the northern Calabria-Lucania Apennines, extrapolated
down to the depth of ~10-12 km. The model includes all the faults playing a significant role, either direct or indirect, on the
seismogenesis of the study area.

-Based on the dimension and shape of the seismogenic patches, an estimate of the maximum expected magnitudes has been
calculated using appropriate scaling relationships, for all the active faults of the Pollino area. The complete rupture of
individual W-dipping faults which are recognized to have been causative of the 2010-2014 seismic sequence are expected to
release a magnitude of ~$M_w$= 6.1 for the VCT and the RSB, and of ~$M_w$= 6.2 for the MPR.  Higher values, up to $M_w$=6.4,
could be reached in the case of the complete and concurrent rupture on both RSB and VCT. The obtained values exceed the
magnitudes of the associate earthquakes which struck the area to date, implying that the aforesaid faults released only partially
their seismogenic potential.

- The delimitation of the fault patches broken during the 2010-2014 seismic sequence, and their geometrical parameterization,
allowed us to verify that a high consistency occurs between the theoretical magnitudes based on the Subsurface Rupture Length
and the magnitudes of the mainshocks.
The estimates provided, for the VCT fault (which released the $M_w$ 5.2 Mormanno earthquake) a $M_w$ 5.3 and, for the MPR fault
(which released the $M_w$ 4.0 and 4.3 Morano Calabro earthquakes), a $M_w$=4.5. The magnitudes calculated using the relationships
based on the Subsurface Rupture Area (4.9<$M_w$<5.0 for the VCT and 4.5<$M_w$<4.6 for the MPR), instead deviate more from
the observed values.

- Our reconstruction confirms that the western segment of the Pollino Fault, despite not being presently active, seems to
maintain a significant seismotectonic role. In fact, juxtaposing crustal sectors with different structure and composition
(Apennine platform domain to the north, and San Donato metamorphic core to the south) acts as a barrier to the southern
propagation of the seismogenic faults of the Mercure-Campotenese sector (ROCS, MPR), limiting their dimensions and
seismogenic potential.



In conclusion, we want to point out that also in the case of moderate-to-minor seismic sequences, as the Pollino 2010-2014
one, the approach based on the three-dimensional reconstruction of the directly involved, as well as neighboring, Quaternary
fault surfaces, represents a real breakthrough in the seismotectonic analysis and, ultimately, in the cognitive path that leads to
a better assessment of the seismic hazard of an active area.

**Author contribution**:
DC, FB conceived and conducted the study. FB, DC, FF, SB Wrote the manuscript. DC developed the 3D model. DC, SB, FF
did GIS analysis and mapping. DC, FB, SB performed the fieldwork. CT, DP, BO, RdN, led the seismological analysis. CT,
DP, BO, processed seismological data. FF did geological and seismological stress field inversion. DC prepared the figures.
GL, SB, FB reviewed the figures. DC, SB prepared the GIS geological database. All authors reviewed the final version of the
manuscript.

**Competing interests**: The authors declare no conflict of interest.

**Acknowledgements:**
Funding was from the DPC-INGV PROJECTS-S1 2014-2015 URUnich, resp. F. Brozzetti and from DiSPUTer Department
research funds to G. Lavecchia and F. Brozzetti. This research was supported by PRIN 2017 (2017KT2MKE) funds from the
Italian Ministry of Education, University and Research (P.I. Giusy Lavecchia). The author is grateful to Petroleum Experts,
who provided them with the Move, 2019.1 suite software licence.

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
