# Peer review of "Structural complexities and tectonic barriers controlling recent 1 seismic activity in the Pollino area (Calabria-Lucania, Southern Italy) 2 - constraints from stress inversion and 3D fault model building. 3"

_Solid Earth, 2021_

## Author Response (AR1)

**Chieti, November 13, 2021**

Dear Editor,

Attached please find the revised version of the manuscript entitled "***Structural complexities and tectonic barriers controlling recent seismic activity in the Pollino area (Calabria-Lucania, Southern Italy) - constraints from stress inversion and 3D fault model building, by*** Daniele Cirillo, Cristina Totaro, Giusy Lavecchia, Barbara Orecchio, Rita de Nardis, Debora Presti, Federica Ferrarini, Simone Bello and Francesco Brozzetti, submitted for consideration to the journal "Solid Earth".

The comments and suggestions we received from the two referees were very helpful. We accepted the hints aimed at improving the paper and we did our best to properly modify the manuscript in order to follow them.

All the major changes and new parts of the manuscript are indicated in the response letter and in the attached tracked version of the revised manuscript, in which all the added and/or modified sentences appear in red.

The organization of the text and the English language were carefully reviewed and edited.

We trust we have answered properly the reviewers' remarks and that the revised manuscript is now suitable for publication in the journal Solid Earth.

Best regards,

PhD **Daniele Cirillo**

**Dear Editor of the journal "Solid Earth"**

During our revision process, we shared most of the reviewers' comments so, basically, we accepted all their requests for revision and did our best to improve the quality of the manuscript.

To allow a check of the additional work that we have made, in this letter we have replied to the major points raised by the Reviewers (reported in red) and listed the main changes to the text.

All the changes we have made are shown in the tracked manuscript.

Replies to REV1 Comments

*1 - "the manuscript is highly fragmented, and the unified scientific overall story and implication are quite weak"*

We have made an effort to comply with the reviewer's request and in particular:

- we have better explained in the "Introduction" section the scopes of our work and the workflow, highlighting the connections among its various parts. In particular, we have better integrated the structural constraints deriving from the surface data with those coming from the seismology.
  We have better defined the role of the geological and seismological tensorial analysis, and better integrated it with the field and seismological data. We believe that it is important to show the consistency between the stress fields obtained from both the datasets (structural data and earthquake focal mechanisms), and to confirm the good agreement between the quaternary extensional deformations surveyed at the surface and the coseismic deformations recorded in the area;
- to further improve understanding of the "scientific story" the description of the seismological data has been moved immediately after the description of geological data and before discussing the inversion of the stress field, which uses both geological and seismological data;
- for the same reason as above, we moved figures 2 and 3 (geological data and focal pseudo-mechanisms) to the section "Structural survey and fault kinematic analysis";

*2 - "there are lots of inconsistent explanations with different scales that are hard to understand quantitatively"*

In the revised text, we met this request by reorganizing significantly and partially re-writing, the "Data and Methods" and the "Discussion" sections.
We also better explained the relationships occurring between the different sets of active faults depicted in the 3D structural model, and the 2010-2014 seismicity, to make the subsurface reconstruction of the seismogenic structures more convincing.
We have also illustrated more in-depth the parameterization of the active faults and clarified some important features, as f.i. the "seismogenic patches" that ruptured during the seismic sequence. We stressed the reasoning that the comparison between the expected magnitude (obtained from the surface fault parameters) with the estimates of the values (based on the size of the seismogenic patch) could have great importance to ascertain the future seismogenic potential of the seismogenic structures.
The time-space relationships between the fault system activated by the recent seismicity and the Pollino fault, which has a long tectonic history and shows a very different strike, have been better described to hypothesize the role of tectonic-barrier played by this latter fault in the propagation of the coseismic ruptures.

*3 - "I think if the following sections of abstract/introduction/discussions/conclusions are written more succinctly and significantly improved in order to let the reader get all the salient facts, and I would have no problem in recommending publication"*
To comply with this request of Rev1 we have combined different chapters in just one, for a better comprehension of the text, and to improve the logical thread from different scales.
In particular, we combined the "3D Fault Model Building" section (which was previously included in the "Data and Methods") with the "3D Fault Model Building" chapter which was too methodological to be among the results. The respective figure follows the text.
In the "Results", however, remains "3D Fault Model of the Pollino area fault system" which contains results.
This change and other similar ones allowed us to significantly reduce the text (about 15-20%).
We added, in the supplementary material, the "Acronym list" for easier reading (suppl. Text 1)
Moreover, we have made an accurate revision of the English writing

Replies to REV2 Comments

*1 - "The paper is in general well written even if a general reorganization of the paragraph is needed to follow better the text*
A very similar comment had been made by the REV1. we think we have answered in detail this point with replies 1 and 3 to REV1.

*2 – "What I mean is to separate literature data from new results, and the latter from interpretation. For instance, I suggest to not include section 2.1 (Geological Setting) in section 2 but to separate them, and I invite the authors to continue with the same spirit"*
As regards the Geological setting section, we made the requested change and separated it from the Seismotectonic Setting section.
Further, we reorganized the sections dealing with the structural-geological data and the seismological ones, and only after describing both of them, we present the tensor analysis to compare the Geological and seismological stress fields.
in general, in the new reorganization of the manuscript we have been particularly careful to separate the previous literature data, the results of our original elaborations from the interpretations and speculative considerations
Other significant revisions that meet this request of REV2, are those explained in our previous reply to point 3 of REV1

*3 – the number of sections should be reduced, and a review of English is necessary.*
Overall, we could not reduce the number of sections but thanks to the significant reorganization of the text and the careful revision, also of the English language, it was possible to sensibly reduce its length (see also reply to point 3 of REV1)

---

## Author Response (AR2)

Dear Editor,

We want to thank you for your comments.

We addressed the points highlighted and provide here a point-by-point response.

Furthermore, we provide a track-changes version of the manuscript and a version with changes incorporated.

Best regards

Daniele Cirillo on behalf of the co-authors.

- Line 31: there are so many other seismotectonic studies that use subsurface and field data to reconstruct 3D fault geometry, as the recent papers by Barchi et al., 2021, Ross et al., 2020 and Porreca et al., 2020.

Done

- Line 32: remove ";" SCEC, 2021

Done

- Line 53: "at depth"

Done

- Line 64: you should cite also other groups working on seismotectonic studies: Sato et al., 1998 Kobe; Bonini et al. 2014 Emilia; Gracia et al., 2019 Morocco; Barchi et al., 2021; Porreca et al., 2018 Vettore.

Thanks, we added them

- Line 66: "," after the parenthesis

Done

- Line 68: you state: "In the sector", but it is not clear which sector. Please specify it better.

Done

- Line 69: please delete "time interval"

Done

- Fig. 1b. Replace "Lucania" with "Basilicata". I think Basilicata is more currently used than Lucania.

Done

- Lines 111-112: refer also to the group of Mattei et al., 2007; Cifelli et al., 2007

Done

- Line 220: delete the space before the ")"

Done

- figure 4 is too small: it is difficult to distinguish the color of the events. Please, try to arrange the figure in a square with 4 panels.

we have modified the figure as suggested

- Line 259: what is F5?

Thanks, we specified it in the main text

- Line 260: "and ii) to favor"

Done

- Fig. 5: The principal stress data are reported in (b) as dip and azimuth (opposite with respect to (a)). Sigma-1 in (b) has not the same value as reported in the Table c. Please check all the data.

we have modified as suggested

- Line 351: please add references after "since Middle Pleistocene"

Done

- Line 365: replace "lengthened" with "oriented"

Done

- Line 367: if Cluster 1 started in 2011, why do you state that it lasted "for the entire 2010-2014 seismic activity"?

Thanks for pointing out this issue. It was a typos.

- Line 387: replace with "The obtained 3DFM"

Done

- Line 427: the extension of the Apennine chain is quantitatively assessed not only by borehole data but also from geodetic data. Please add some references (e.g. D'Agostino et al., 2001).

Done

---

## Author Response (AR3)

**Comments to the author**:
Dear Authors,
Based on the Topical Editor's report and overall assessment of the review process, I am pleased to accept your manuscript for publication in Solid Earth pending the few technical corrections as outlined by the Guest Editor.

Many thanks for choosing Solid Earth as a platform to publish your research.

Best regards
Federico Rossetti

Dear Executive Editor,

Thank you for your email. In the new version of the manuscript, we have accepted all the suggestions highlighted by the Guest Editor and made the technical corrections.

Many thanks for accepting our paper in the journal of Solid Earth.

On behalf of all co-author

Best Regard

Daniele Cirillo

**Comments to the author**:

Dear authors,

I'm happy to announce that your paper can be published in the Special Issue of Solid Earth journal.

Please, take note of these minor typos during the draft production:

- line 15: "and, at depth, by using..."

- line 21: replace "activity" with "seismicity"

- line 63: delete "," before "seismogenic faults"

- line 131: delete ")" before Cheloni et al., 2017)

- line 354: Brozzetti et al., 2017a

We have corrected all the typos above highlighted during the draft production.

Best regards,
M. Porreca

Dear Guest Editor,

Thank you for your attention and for your suggestion about the revision of our manuscript.

We are pleased that our work has been accepted in Solid Earth.

On behalf of all co-author

Kind Regard

Daniele Cirillo